

# Reliability and utility of blood glucose levels in the periodontal pockets of patients with type 2 diabetes mellitus: a cross-sectional study

Yutaka Terada[1], Hiroyuki Watanabe[1], Mari Mori[1,2], Kotoko Tomino[1], Masaya Yamamoto[1], Mitsuru Moriya[3,4], Masahiro Tsuji[5], Yasushi Furuichi[1,6,7], Tomofumi Kawakami[1,4] and Toshiyuki Nagasawa[1,6,8]

[1] Division of General Dentistry, Health Sciences University of Hokkaido Hospital, Sapporo, Hokkaido, Japan
[2] Division of General Dental Sciences, Department of Oral Rehabilitation, School of Dentistry, Health Sciences University of Hokkaido, Sapporo, Hokkaido, Japan
[3] Division of Internal Medicine, Psychosomatic Internal Medicine, Health Sciences University of Hokkaido Hospital, Sapporo, Hokkaido, Japan
[4] Institute of Preventive Medical Science, Health Sciences University of Hokkaido, Sapporo, Hokkaido, Japan
[5] Division of Diabetes and Metabolism, Tenshi Hospital, Sapporo, Hokkaido, Japan
[6] Division of Periodontology and Endodontology, Department of Oral Rehabilitation, School of Dentistry, Health Sciences University of Hokkaido, Ishikari-Tobetsu, Hokkaido, Japan
[7] Division of Dental Education Development, School of Dentistry, Health Sciences University of Hokkaido, Ishikari-Tobetsu, Hokkaido, Japan
[8] Division of Advanced Clinical Education, Department of Integrated Dental Education, School of Dentistry, Health Sciences University of Hokkaido, Ishikari-Tobetsu, Hokkaido, Japan

Corresponding author
Toshiyuki Nagasawa,
nagasawa@hoku-iryo-u.ac.jp

## ABSTRACT

**Background:** Several studies have measured gingival blood glucose (GBG) levels, but few have confirmed systematic bias using Bland–Altman analysis. This study compared the effectiveness of GBG levels with that of fingertip blood glucose (FTBG) levels using Bland–Altman and receiver operating characteristic (ROC) analyses.

**Methods:** A total of 15 healthy volunteers and 15 patients with type 2 diabetes were selected according to inclusion and exclusion criteria. Each group comprised eight male and seven female participants. The GBG and FTBG levels were measured using a self-monitoring blood glucose device after periodontal examination. Pearson's product-moment correlation and simple linear regression analyses were performed. In addition, Bland–Altman analysis was also performed to assess the degree of agreement between the two methods. ROC analysis was conducted to determine the sensitivity, specificity, and cutoff values for patients with diabetes. The area under the ROC curve (AUC) was used to identify significant differences.

**Results:** The mean GBG and FTBG levels were 120 ± 44.8 mg/dL and 137 ± 45.1, respectively, for the whole sample. The mean GBG and FTBG levels were 145 ± 47.2 mg/dL and 163 ± 49.1, respectively, in the diabetes group. The mean GBG and FTBG levels in the nondiabetes group were 95.3 ± 25.2 and 111 ± 18.8, respectively. Patients with diabetes were more likely to have a probing pocket depth (PPD) of ≥4 mm at the sampled site. Pearson's product-moment correlation and simple linear regression analyses revealed a significant correlation between the GBG and FTBG

measurements. Bland–Altman analysis revealed that GBG and FTBG measurements differed significantly among all participants; however, no significant differences were observed among the patients with diabetes (mean difference (MD) ± standard deviation (SD) = −18.1 ± 34.2, 95% confidence interval (CI) [−37.0 to 0.88]) or among the participants with a PPD of ≥4 mm (MD ± SD = −15.2 ± 30.4, 95% CI [−30.8 to 0.43]). The sensitivity, specificity, and cutoff values of the GBG measurements for detecting diabetes were 80%, 93%, and 123.5 mg/dL, respectively. The sensitivity, specificity, and cutoff values of the FTBG measurements for detecting diabetes were 73%, 87%, and 134.0 mg/dL, respectively. No significant differences were observed between the AUCs (0.078, 95% CI [−0.006 to 0.161]).

**Conclusions:** The GBG measurements aligned with the FTBG measurements in the patients with diabetes and among the participants with a PPD of ≥4 mm. Patients with diabetes were more likely to have a PPD of ≥4 mm at the sampled site, GBG levels can be used to screen for type 2 diabetes in dental clinics.

## INTRODUCTION

Type 2 diabetes mellitus, characterized by chronic hyperglycemia, can cause complications if not properly treated. Moreover, there is a bidirectional relationship between periodontal disease and diabetes. Patients with diabetes are at high risk of developing periodontal disease and its progression, which in turn affects the progression of diabetes mellitus. Periodontal disease also affects the progression of diabetes mellitus (*Graziani et al., 2018*). Appropriate periodontal treatment plays an important role in managing patients with diabetes, as periodontal treatment improves insulin resistance and blood glucose management (*Simpson et al., 2022*). The guidelines for periodontal surgery and tooth extraction recommend that hemoglobin (Hb) A1c levels be maintained at 6.9% for patients with diabetes (*Japanese Society of Periodontology, 2014*). Furthermore, dental treatment must be postponed if the blood glucose levels are >200 or <70 mg/dL (*Little et al., 2017*). However, measuring blood glucose levels using venous blood sampling is difficult in dental clinics; therefore, dental treatment commences based on physician information. Given the recent relationship between periodontal disease and diabetes, the inability to measure blood glucose levels in the dental outpatient setting is a major problem. Bleeding from gingiva can be a promising resource for measuring blood glucose for dentists as it would be less burden on the patient. If gingival blood glucose measurements are consistent with those of the fingertip, blood glucose measurement from the gingiva can be incorporated into dental blood glucose measurement.

Previous studies on gingival blood glucose (GBG) measurements concluded that GBG levels could be used for diabetes screening (*Stein & Nebbia, 1969*; *Tsutsui, Rich & Schonfeld, 1985*). A self-monitoring blood glucose (SMBG) device enables patients to

measure their glucose levels by puncturing their fingertips. Several studies have reported strong correlations between fingertip blood glucose (FTBG) levels acquired using an SMBG device and GBG levels (*Parker et al., 1993*; *Beikler et al., 2002*; *Khader et al., 2006*; *Ardakani et al., 2009*). The measurement of GBG levels is a rapid, safe, noninvasive screening method for diabetes that can be performed during routine periodontal examinations (*Parker et al., 1993*; *Beikler et al., 2002*; *Khader et al., 2006*; *Ardakani et al., 2009*). Additionally, more patients with diabetes preferred measuring GBG levels over measuring FTBG (*Rosedale & Strauss, 2012*).

The intraclass correlation coefficient is often used as a measure of reliability to validate new measurement measures. However, what is important in actual measurement is the degree of disagreement. When introducing a new measure, it is necessary to know how much it may differ from previous measures. Since the acceptable measurement error in a clinical setting varies among the types of measurements, no standard can be set automatically by statistical methods (*Müller & Behbehani, 2005*). Therefore, *Bland & Altman (1986*, *1999)* reported that an agreement analysis between the two methods was needed.

The Bland–Altman analysis has been used to clarify systematic bias (*Bland & Altman, 1986*, *1999*). Only two previous studies have reported Bland–Altman analyses of GBG and FTBG levels (*Müller & Behbehani, 2004*, *2005*; *Strauss et al., 2009*). Half of the 46 participants in one group did not have periodontitis in the study by *Müller & Behbehani (2004*, *2005)* and only 15% had diabetes. Moreover, Bland–Altman analysis of the GBG and FTBG measurements revealed low concordance between the GBG and FTBG levels (*Müller & Behbehani, 2005*). The severity of periodontitis was unknown in the study by *Strauss et al. (2009)* and only 9% of the participants had diabetes. Bland–Altman analysis revealed adequate agreement between the GBG and FTBG levels in this study (*Strauss et al., 2009*). Thus, the Bland–Altman analysis results of GBG and FTBG levels have been inconsistent.

To date, several studies have reported that GBG and FTBG values are associated with diabetes and useful for diabetes screening (*Suneetha & Rambabu, 2012*; *Gaikwad et al., 2013*; *Shetty et al., 2013*; *Kaur, Singh & Sharma, 2013*; *Gupta et al., 2014*; *Dwivedi et al., 2014*; *Shylaja et al., 2016*; *Rajesh et al., 2016*; *Parihar et al., 2016*; *Partheeban et al., 2017*; *Sibyl et al., 2017*; *Mirza et al., 2018*; *Sande et al., 2020*; *Rapone et al., 2020*; *Wu et al., 2021*; *Patel et al., 2023*; *Dash et al., 2023*; *Alqazlan et al., 2024*). Alternatively, other researchers reported that GBG and FTBG values are not associated with diabetes and are not useful for diabetes screening (*Debnath et al., 2015*; *Ansari Moghadam et al., 2024*). A recently published systematic review and meta-analysis (*Fakheran et al., 2024*), citing the three articles mentioned above (*Müller & Behbehani, 2004*, *2005*; *Strauss et al., 2009*), reported that the GBG values are useful if gingival inflammation is strong and bleeding on probing (BOP) is high.

This study aimed to evaluate the utility of GBG measurements compared with that of FTBG measurements *via* Bland–Altman and receiver operating characteristic (ROC) analyses by determining the GBG and FTBG levels in patients with and without diabetes

who visited the Division of General Dentistry, Health Sciences University of Hokkaido Hospital (Fig. 1).

## MATERIALS AND METHODS

### Sample size setting

Previous studies were extracted from the PubMed database and examined to determine the number of participants required to evaluate the efficacy of GBG measurements compared with that of FTBG measurements. One study (*Rajesh et al., 2016*) had 24 participants, one study (*Gaikwad et al., 2013*) had 25 participants, and three studies (*Gupta et al., 2014*; *Shylaja et al., 2016*; *Sibyl et al., 2017*) with 30 participants reported a significant correlation between the GBG and FTBG measurements. G*Power 3.1 (*Faul et al., 2007*) obtained from an internet website (*The G*Power Team, 2007*) was used to calculate the sample size. The $\alpha$ and $\beta$ errors were set to 0.05 and 0.2, respectively. According to a previous report, a correlation coefficient of 0.715 was selected for the correlation analysis (*Gupta et al., 2014*). The effect size was set as 0.8 for paired t tests evaluated *via* Bland–Altman analysis. The sample sizes were calculated as 12 for the correlation analysis and 15 for the paired t test using G*Power 3.1. The total number of participants was 30, with 15 participants each in the diabetic and nondiabetic groups.

Inclusion criteria

1. Twenty years or older with at least one current tooth undergoing dental treatment at the Division of General Dentistry, Health Sciences University of Hokkaido Hospital.

2. We classified patients with diabetes based on the specialists' diabetes diagnoses. Patients in the diabetes group underwent treatment at the Division of Internal Medicine, Health Sciences University of Hokkaido Hospital.

Exclusion criteria

1. Pregnant or lactating women.

2. Patients without diabetes with GBG or FTBG levels greater than 200 mg/dL.

This study adhered to the tenets of the Declaration of Helsinki. Written informed consent was obtained from the participants after the study was explained. This study was approved by the Ethics Committee of the Institute of Preventive Medical Science, Health Sciences University of Hokkaido (No. 2019_028).

### Medical and dental examinations

Details regarding sex, age, the time at which the participants finished their last meal on the day of examination, whether they had visited a physician, and smoking history were recorded. In addition, the most recent HbA1c levels were also recorded, and patients with type 2 diabetes were assigned to the diabetes group.

Multiple examiners conducted periodontal examinations, and an interexaminer reliability assessment was conducted. Dental examinations were subsequently performed. The number of teeth present was recorded. In addition, the dental plaque was stained, and the plaque control record (PCR) was determined (*O'Leary, Drake & Naylor, 1972*). The probing pocket depth (PPD) and BOP were evaluated using a periodontal pocket probe (CP-11; Hu-Friedy, Chicago, IL, USA) and recorded. Tooth mobility was measured with

**Fingertip blood glucose (FTBG)**

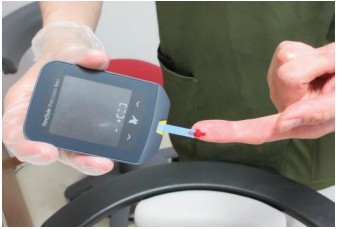

**Participants**
**15 type 2 diabetes mellitus (Male 8, Female 7)**
**15 non diabetes mellitus (Male 8, Female 7)**

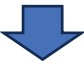

**Measurement**
**Blood collection: Fingertip (FT), Gingival blood (GB)**
**Blood glucose measurement: Self-monitoring blood glucose device**
**Measurement of probing pocket depth (PPD) at the sampling site**

**Gingival blood glucose (GBG)**

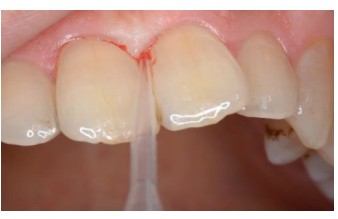

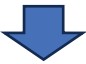

**Analysis**
**Agreement with FTBG: the Bland-Altman analysis**
**Distinction between diabetics and nondiabetics: ROC analysis**

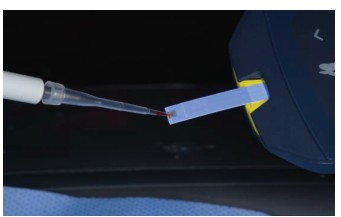

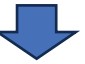

**Results**
The Bland Altman analysis: no significant differences between GBG and FTBG among the patients with diabetes or the participants with a PPD ≥4 mm.

The sensitivity, specificity, and cutoff value of the GBG level measurements for detecting diabetes were 80%, 93%, and 123.5 mg/dl, respectively.

**Figure 1 Flow chart indicating the outline of this study.** Participants were type 2 diabetics and non-diabetics. Fingertip (FT) and gingival (G) blood glucose (BG) levels were measured using a self-monitoring blood glucose device. The Bland-Altman analysis was used to test agreement with FTBG, and ROC analysis was used to distinguish diabetic and nondiabetic patients. The person in all images was the first author (Dr. Yutaka Terada). This figure has been drawn using PowerPoint 2019.

dental tweezers using Miller's mobility index (*Japanese Society of Periodontology, 2017*; *Wu et al., 2018*) and classified as follows: Grade 0 (physiological mobility), ≤0.2 mm; Grade 1 (slight, labiolingual), 0.2–1 mm; Grade 2 (moderate, labiolingual and mesiodistal), 1–2 mm; and Grade 3 (severe, labiolingual and mesiodistal) ≥2 mm or vertical movement. The periodontal inflamed surface area (PISA) and periodontal epithelial surface area (PESA) were calculated (*Nesse et al., 2008*) using a spreadsheet (*Vissink et al., 2008a*) available on an internet website (*Vissink et al., 2008b*), and the PPD and BOP were entered. Periodontitis was diagnosed and classified based on the examinations and radiographs (*Tonetti, Greenwell & Kornman, 2018a*, *2018b*). Stages indicating the severity and complexity of periodontitis were as follows: Stage I, initial; Stage II, moderate; Stage III, severe with potential for additional tooth loss; and Stage IV, severe with potential for loss of dentition. The extent was defined as generalized (>30% of teeth involved) or localized for each stage. In addition, the grade, which indicates the risk of periodontitis progression, was defined as follows: Grade A, slow rate of progression; Grade B, moderate rate of progression; and Grade C, rapid rate of progression. Risk factors, including smoking and diabetes, were considered when the grade was determined.

## Blood glucose measurements

The dental plaque was gently removed with cotton pellets, and the sampling sites were isolated using cotton rolls to prevent contamination with saliva and dental plaque. A saliva ejector was also used. Periodontal probing of the sampling sites was subsequently performed. A micropipette (Eppendorf Reference 2, Eppendorf AG, Hamburg, Germany) collected approximately 1.0 µL (the minimum volume required for a single blood glucose level measurement) of blood from the sampling site. The mandibular sampling sites were restricted to the labial or buccal sides of the teeth to prevent salivary contamination. The labial or palatal sides of the maxillary anterior teeth were sampled, whereas the palatal sides of the maxillary premolars or molars were sampled. Periodontal pockets with pus were excluded. Blood samples were also acquired from the fingertip. The blood glucose levels were measured using an SMBG device (FreeStyle Precision Neo; Abbott Diabetes Care, Inc., Alameda, CA, USA) immediately after blood collection.

## Statistical analysis

### Clinical characteristics of the participants

All the statistical analyses were performed *via* SPSS Statistics Version 26 (IBM, Chicago, IL, USA). Categorical variables of the diabetic and nondiabetic groups were analyzed according to sex, smoking habits, diagnosis and classification of periodontitis, and PPDs of ≥4 and ≤3 mm at the blood sampling site. The chi-square or Fisher's exact tests were used to examine these parameters. Age and HbA1c levels in the diabetic group, blood glucose levels, and postprandial time were continuous variables. The continuous dental variables were the number of teeth present, PPD, BOP, PISA, PESA, PCR, and tooth mobility. The mean ± standard deviation (SD) values were calculated. Continuous variables of the diabetic and nondiabetic groups were analyzed using a two-sample t tests.

### Correlation and regression analyses

The correlation between the GBG and FTBG measurements was analyzed to determine whether the GBG measurements were as reliable as the FTBG measurements. The Pearson's product-moment correlation coefficient (r) was used to evaluate the correlation. In addition, the simple linear regression equation {FTBG = constant + simple linear regression coefficient (R) × GBG} and the coefficient of determination ($R^2$) for simple linear regression analysis were determined (*Müller & Behbehani, 2005*; *Rajesh et al., 2016*). In the case of simple linear regression analysis, there is only one explanatory variable, so the R is equal to the r.

### Bland–altman analysis

The Bland–Altman analysis was performed as follows (*Bland & Altman, 1986*, *1999*; *Müller & Behbehani, 2005*; *Strauss et al., 2009*; *IBM Support, 2020*). The interpretation of bias was based on the mean difference (MD) between the GBG and FTBG values. Bias and MD are synonymous. The limit of agreement (LOA) is the range corresponding to bias (MD) ± 1.96 × SD for the 95% confidence interval (CI), and theoretically, 95% of the differences in the measured values converge to the LOA. The MD ± SD between GBG and FTBG and the 95% confidence interval (95% CI) of the MD were calculated. A one-sample

t test was then used to compare GBG and FTBG, and a fixed bias was considered present if the 95% CI did not exceed zero. The coefficient of agreement (COA) and LOAs were determined. The COA was calculated as $1.96 \times$ SD. The LOA was calculated as the MD $\pm$ COA. The 95% CIs of the upper and lower LOAs were also calculated. A simple linear regression analysis was performed with the difference between GBG and FTBG as the objective variable and (GBG + FTBG)/2 as the explanatory variable to determine the presence of proportional bias, and the significance was tested. Proportional bias was considered present if the R was judged to be significant. Furthermore, only the MD $\pm$ SD and evaluation up to 95% CIs were used to determine fixed bias in the presence of fixed and proportional bias. Sites with a PPD of $\geq 4$ mm were considered suitable for obtaining the same GBG value as the FTBG value (*Strauss et al., 2009*). A subgroup analysis for PPDs of $\geq 4$ and $\leq 3$ mm was also conducted.

### Receiver operating characteristic analysis

The ROC curves of the GBG and FTBG levels were plotted, and the areas under the curves (AUCs) were calculated. The sensitivity and specificity were also calculated. The optimal cutoff values for diabetes screening were subsequently determined. The cutoff values were calculated as the coordinate point, *i.e.*, the minimum value of $(1\text{-sensitivity})^2 + (1\text{-specificity})^2$ from the coordinate point table of the ROC curve (*Strauss et al., 2012*, *2015*). The difference between the AUCs was determined *via* a one-sample t test if the AUCs of the GBG and FTBG values were significant. All significance levels were set at $p < 0.05$.

## RESULTS

Informed consent was obtained from 37 participants between November 2020 and June 2021. The diabetic and nondiabetic groups comprised 15 and 22 participants, respectively. Therefore, case–control matching was performed in the diabetic and nondiabetic groups for 15 participants. Eight male and seven female participants matched perfectly. The analysis was conducted under these conditions, as the minimum age difference possible was within 8 years. The Supplemental File (Data S1) presents the raw data before analysis.

The diabetic group did not include patients with Grade A periodontitis; in contrast, 14 of the 15 participants (93%) in the nondiabetic group had Grade A or B periodontitis (Table 1), indicating a significant difference ($p = 0.001$). The participants in the diabetic group were significantly more likely to have generalized chronic periodontitis (14 of the 15 patients, 93%) ($p = 0.040$). The mean number of present teeth was $19.9 \pm 4.85$ and $23.7 \pm 3.81$ in the diabetic and nondiabetic groups, respectively, indicating a significant difference ($p = 0.026$; Table 2).

Figure 2 shows the correlation and simple linear regression between the GBG and FTBG levels in all participants. The correlation coefficient and the simple linear regression equation were r = 0.827 ($p < 0.001$) and FTBG value = 36.970 + 0.827 × GBG value (95% CI for R [0.613–1.051], $p < 0.001$), $R^2 = 0.684$, respectively.

The Bland–Altman analysis revealed a significant difference in fixed bias (MD $\pm$ SD = −16.8 $\pm$ 26.4 mg/dL, 95% CI for MD [−26.7 to −6.94 mg/dL], $p = 0.002$; Fig. 3 and Table 3). The LOA ranged from −68.6 to 35.0 mg/dL. Proportional bias was considered nil,

**Table 1  Categorical variables of the participants.**

|  | Non diabetes mellitus $n$ (%) | Diabetes mellitus $n$ (%) | $p$ value |
|---|---|---|---|
| Sex |  |  | NA |
| Female | 7 (23.3) | 7 (23.3) |  |
| Male | 8 (26.7) | 8 (26.7) |  |
| Smoking habits |  |  | 0.299[†] |
| Non-smoker | 14 (46.7) | 12 (40.0) |  |
| Smoker | 1 (3.3) | 3 (10.0) |  |
| Periodontitis stage |  |  | 0.605[†] |
| I | 1 (3.3) | 0 (0.0) |  |
| II | 4 (13.3) | 6 (20.0) |  |
| III | 7 (46.7) | 8 (53.3) |  |
| IV | 3 (10.0) | 1 (3.3) |  |
| Periodontitis grade |  |  | 0.001[†] |
| A | 8 (26.7) | 0 (0.0) |  |
| B | 6 (20.0) | 9 (30.0) |  |
| C | 1 (3.3) | 6 (20.0) |  |
| Periodontitis extent |  |  | 0.040[†] |
| Localized | 6 (20.0) | 1 (3.3) |  |
| Generalized | 9 (30.0) | 14 (46.7) |  |
| PPD of the sampled site |  |  | 0.010[‡] |
| ≤3 mm | 10 (33.3) | 3 (10.0) |  |
| ≥4 mm | 5 (16.7) | 12 (40.0) |  |

Notes:
[†] Fisher's exact test
[‡] chi-square test.
$n$, number; NA, not applicable; PPD, probing pocket depth.

as no significant difference was observed (R = −0.006, 95% CI for R [−0.244 to 0.232], $p$ = 0.959).

Figure 4 presents the correlation and simple linear regression between the GBG and FTBG levels in the nondiabetic group. The correlation coefficient and the simple linear regression equation were r = 0.756 ($p$ < 0.001) and FTBG value = 57.028 + 0.756 × GBG value (95% CI for R = 0.272 to 0.858, $p$ = 0.001), $R^2$ = 0.572, respectively.

The Bland–Altman analysis revealed a significant difference in fixed bias among the participants without diabetes (MD ± SD = −15.5 ± 16.5 mg/dL, 95% CI for MD [−24.7 to −6.41 mg/dL], $p$ = 0.003; Fig. 5 and Table 3). LOA ranged from −47.8 to 16.8 mg/dL. Proportional bias was considered nil, as no significant difference was observed (R = 0.329, 95% CI for R [−0.107 to 0.765], $p$ = 0.127).

Figure 6 depicts the correlation and simple linear regression between the GBG and FTBG levels in the diabetic group. The correlation coefficient and the simple linear regression equation were r = 0.748 ($p$ < 0.010) and FTBG value = 49.966 + 0.748 × GBG value (95% CI for R [0.365–1.194], $p$ = 0.001), $R^2$ = 0.560, respectively.

The Bland–Altman analysis revealed no significant difference in fixed bias among the participants with diabetes (MD ± SD = −18.1 ± 34.2 mg/dL, 95% CI for MD [−37.0 to 0.88
**Table 2 Continuous variables of the participants.**

|  | Total participants (n = 30) | | | Non diabetes mellitus (n = 15) | Diabetes mellitus (n = 15) | |
| --- | --- | --- | --- | --- | --- | --- |
|  | Mean ± SD | Min | Max | Mean ± SD | Mean ± SD | p value |
| Age (years) | 67.1 ± 8.79 | 48 | 85 | 66.8 ± 9.18 | 67.4 ± 8.68 | 0.855 |
| HbA1c (%) | 6.86 ± 0.82 | 5.8 | 9.1 | NA | 6.86 ± 0.82 | NA |
| GBG (mg/dL) | 120 ± 44.8 | 44 | 272 | 95.3 ± 25.2 | 145 ± 47.2 | 0.001 |
| FTBG (mg/dL) | 137 ± 45.1 | 89 | 232 | 111 ± 18.8 | 163 ± 49.1 | 0.001 |
| GBG-FTBG (mg/dL) | −16.8 ± 26.4 | −73 | 47 | −15.5 ± 16.5 | −18.1 ± 34.2 | 0.799 |
| Postprandial time (minutes) | 187 ± 76.2 | 60 | 360 | 203 ± 88.4 | 175 ± 61.7 | 0.324 |
| Number of present teeth | 21.8 ± 4.69 | 9 | 28 | 23.7 ± 3.81 | 19.9 ± 4.85 | 0.026 |
| Mean PPD (mm) | 2.67 ± 1.05 | 1.61 | 5.99 | 2.51 ± 0.94 | 2.84 ± 1.15 | 0.409 |
| PPD of the sampled site (mm) | 4.13 ± 1.80 | 2 | 10 | 3.60 ± 1.96 | 4.67 ± 1.50 | 0.105 |
| BOP (%) | 25.7 ± 22.5 | 2.30 | 88.2 | 20.2 ± 19.1 | 31.2 ± 24.9 | 0.185 |
| PISA (mm²) | 421 ± 634 | 9.4 | 2,721.3 | 335 ± 494 | 508 ± 757 | 0.465 |
| PESA (mm²) | 1,111 ± 567 | 308.4 | 2,997.4 | 1,123 ± 573 | 1,099 ± 581 | 0.910 |
| Number of teeth with PPD ≥6 mm | 2.37 ± 5.41 | 0 | 21 | 1.93 ± 5.43 | 2.80 ± 5.55 | 0.669 |
| Number of teeth with more than grade 2 of Miller's mobility index | 1.40 ± 4.28 | 0 | 22 | 0.87 ± 2.30 | 1.93 ± 5.66 | 0.505 |
| Sites of PPD ≥4 mm (%) | 25.7 ± 29.1 | 0 | 100 | 22.0 ± 26.1 | 29.5 ± 32.4 | 0.493 |
| Sites of PPD ≥5 mm (%) | 15.0 ± 28.0 | 0 | 95.8 | 12.1 ± 25.3 | 17.8 ± 31.1 | 0.585 |
| Sites of PPD ≥6 mm (%) | 11.1 ± 24.8 | 0 | 88.2 | 9.40 ± 22.8 | 12.7 ± 27.4 | 0.722 |
| PCR (%) | 34.5 ± 16.8 | 8.3 | 69.1 | 31.2 ± 16.1 | 37.7 ± 17.4 | 0.297 |

**Notes:**
p value for two-sample t test. n, number; SD, standard deviation; Min, minimum; Max, maximum; HbA1c, hemoglobin A1c; NA, not applicable; GBG, gingival blood glucose; FTBG, fingertip blood glucose; PPD, probing pocket depth; BOP, bleeding on probing; PISA, periodontal inflamed surface area; PESA, periodontal epithelial surface area; PCR, plaque control record.

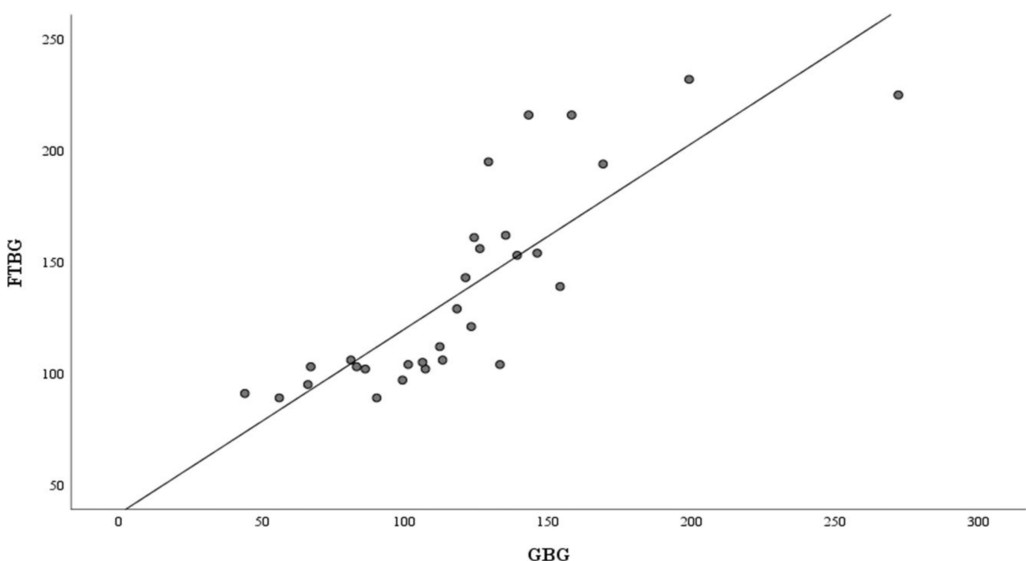

**Figure 2 Correlation and regression analyses for all participants.** Linear regression of GBG (mg/dL) sample measurements on FTBG (mg/dL) sample readings for all participants. The solid line represents the regression line.

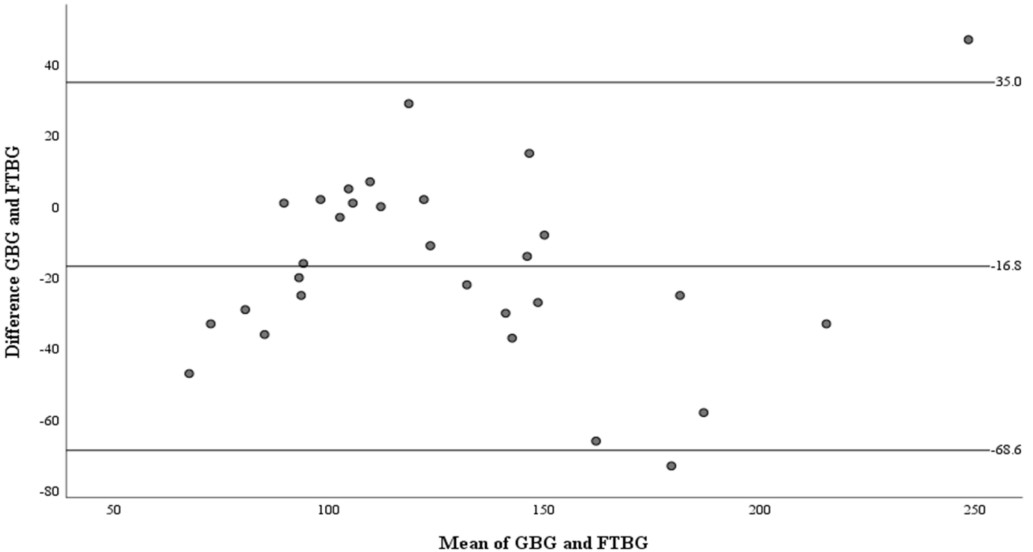

**Figure 3 The Bland-Altman plot for all participants.** Means of GBG (mg/dL) and FTBG (mg/dL) glucose readings and differences in GBG (mg/dL) and FTBG (mg/dL) glucose readings, with the lines representing the upper and lower 95% limits of agreement and the mean of the differences for all participants.

**Table 3 Main analysis of the repeatability/agreement of the measurements of the blood glucose levels (mg/dL) in the GBG and FTBG samples.**

| Measures | Total participants ($n = 30$) | Non-DM ($n = 15$) | Type 2 DM ($n = 15$) |
|---|---|---|---|
| Minimum difference | −73 | −47 | −73 |
| Maximum difference | 47 | 5 | 47 |
| Mean difference ± SD | −16.8 ± 26.4 | −15.5 ± 16.5 | −18.1 ± 34.2 |
| 95% CI of mean difference | −26.7 to −6.94 | −24.7 to −6.41 | −37.0 to 0.88 |
| Coefficient of agreement | 51.8 | 32.3 | 67.0 |
| Limits of agreement | −68.6, 35.0 | −47.8, 16.8 | −85.1, 49.0 |
| 95% CI of the lower limit | −85.7 to −51.5 | −63.6 to −32.0 | −118 to −52.4 |
| 95% CI of the upper limit | 17.9 to 52.1 | 1.00 to 32.6 | 16.3 to 81.7 |

Notes:
 GBG, gingival blood glucose; FTBG, fingertip blood glucose; DM, diabetes mellitus; $n$, number; SD, standard deviation; CI, confidence interval.

mg/dL], $p = 0.060$; Fig. 7 and Table 3). The LOA ranged from −85.1 to 49.0 mg/dL. Proportional bias was considered nil, as no significant difference was observed (R = −0.047, 95% CI for R [−0.501 to 0.408], $p = 0.827$).

Figure 8 depicts the correlation and simple linear regression between the GBG and FTBG levels in the subgroups with a PPD of ≥4 mm at the GBG sampling sites ($n = 17$). The correlation coefficient and the simple linear regression equation were r = 0.838 ($p < 0.001$) and FTBG value = 43.750 + 0.838 × GBG value (95% CI for R [0.499–1.055], $p < 0.001$), $R^2 = 0.572$, respectively, which is significant (Fig. 8).

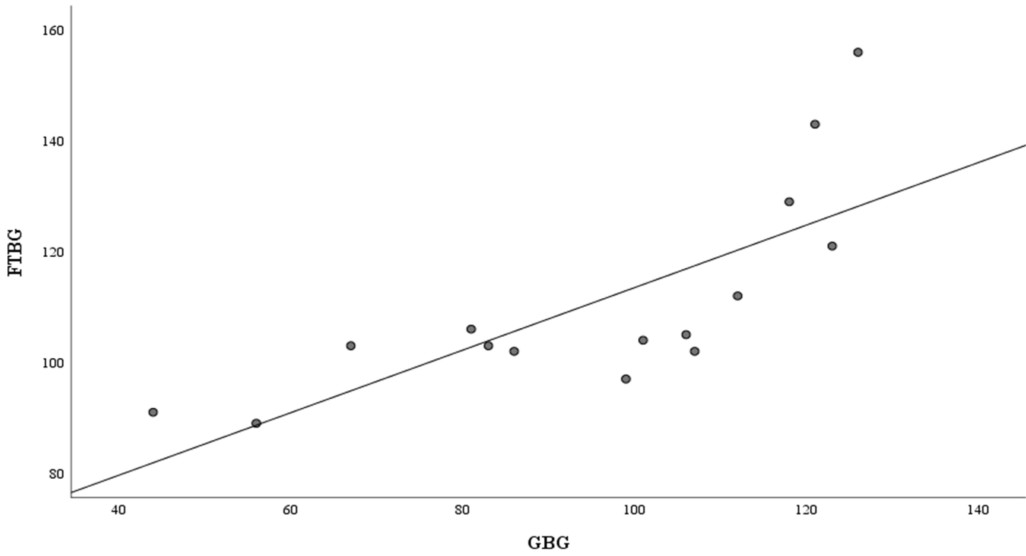

**Figure 4 Correlation and regression analyses for patients without DM.** Linear regression of GBG (mg/dL) sample measurements on FTBG (mg/dL) sample readings for patients without DM. The solid line represents the regression line.

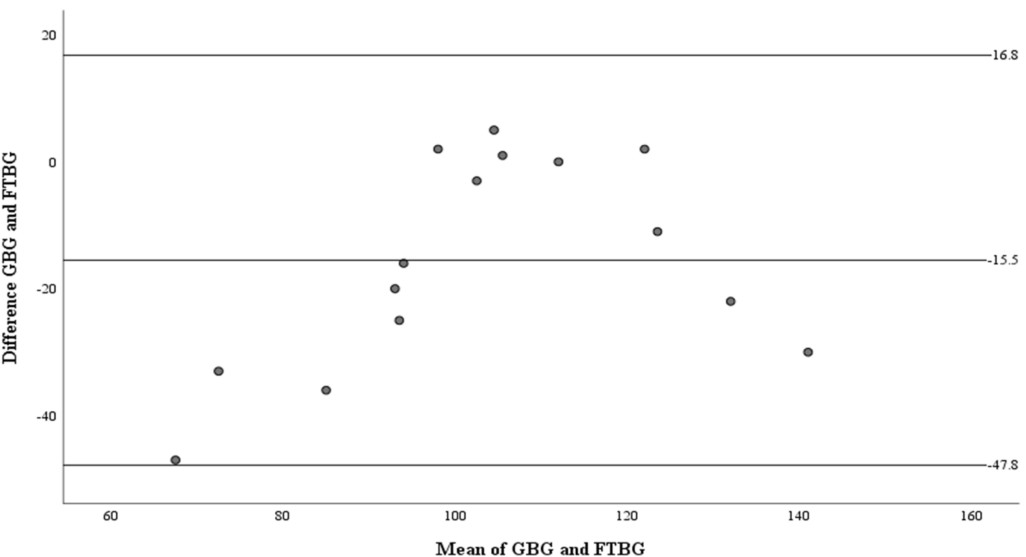

**Figure 5 The Bland-Altman plot for patients without DM.** Means of GBG (mg/dL) and FTBG (mg/dL) glucose readings and differences in GBG (mg/dL) and FTBG (mg/dL) glucose readings, with the lines representing the upper and lower 95% limits of agreement and the mean of the differences for participants without DM.

The Bland–Altman analysis of the PPD ≥4 mm subgroup revealed no significant differences in fixed bias (MD ± SD = −15.2 ± 30.4 mg/dL, 95% CI for MD [−30.8 to 0.43 mg/dL], $p$ = 0.056; Fig. 9 and Table 4). LOA ranged from −74.7 to 44.3 mg/dL. Proportional bias was considered nil, as no significant difference was observed (R = −0.082, 95% CI for R [−0.244 to 0.408], $p$ = 0.599).

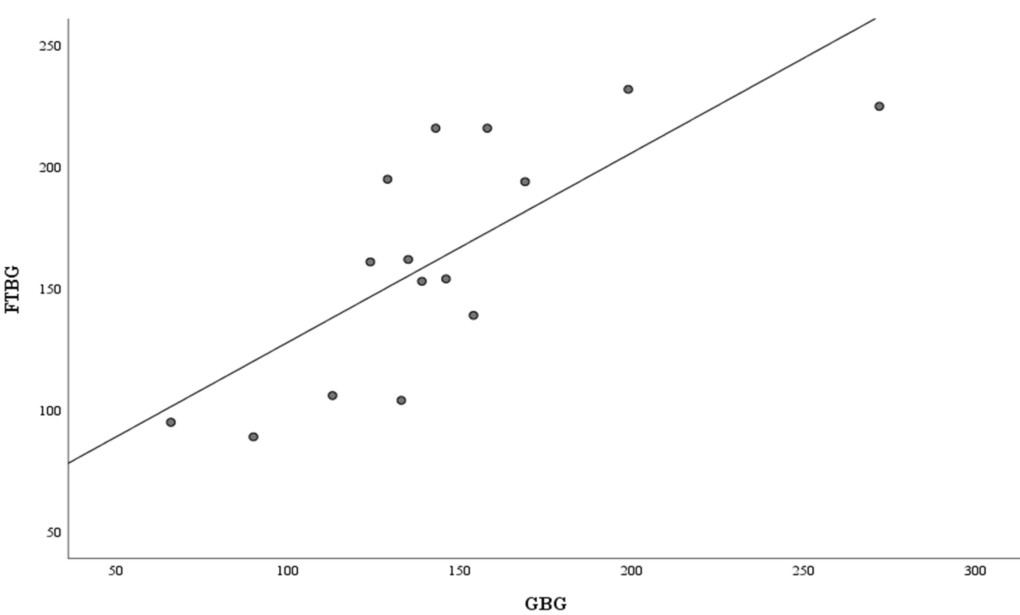

**Figure 6 Correlation and regression analyses for patients with DM.** Linear regression of GBG (mg/dL) sample measurements on FTBG (mg/dL) sample readings for patients with DM. The solid line represents the regression line.               

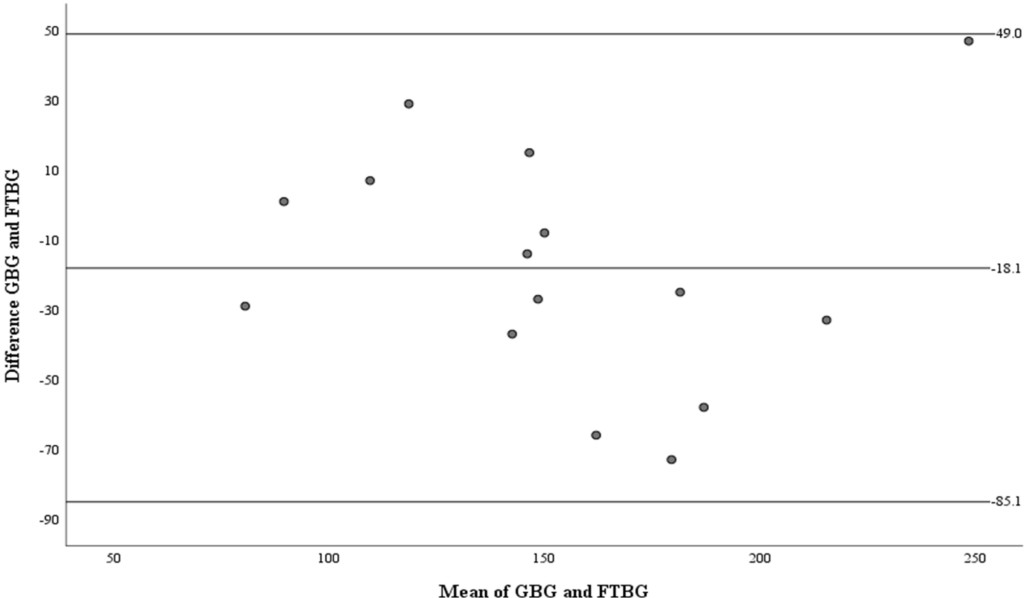

**Figure 7 The Bland-Altman plot for patients with DM.** Means of GBG (mg/dL) and FTBG (mg/dL) glucose readings and differences in GBG (mg/dL) and FTBG (mg/dL) glucose readings, with the lines representing the upper and lower 95% limits of agreement and the mean of the differences for patients with DM.               

Figure 10 depicts the correlation and simple linear regression between the GBG and FTBG levels in the PPD ≤3 mm subgroup ($n$ = 13). The correlation coefficient and the simple linear regression equation were r = 0.823 ($p$ < 0.001) and FTBG value = −2.804 +

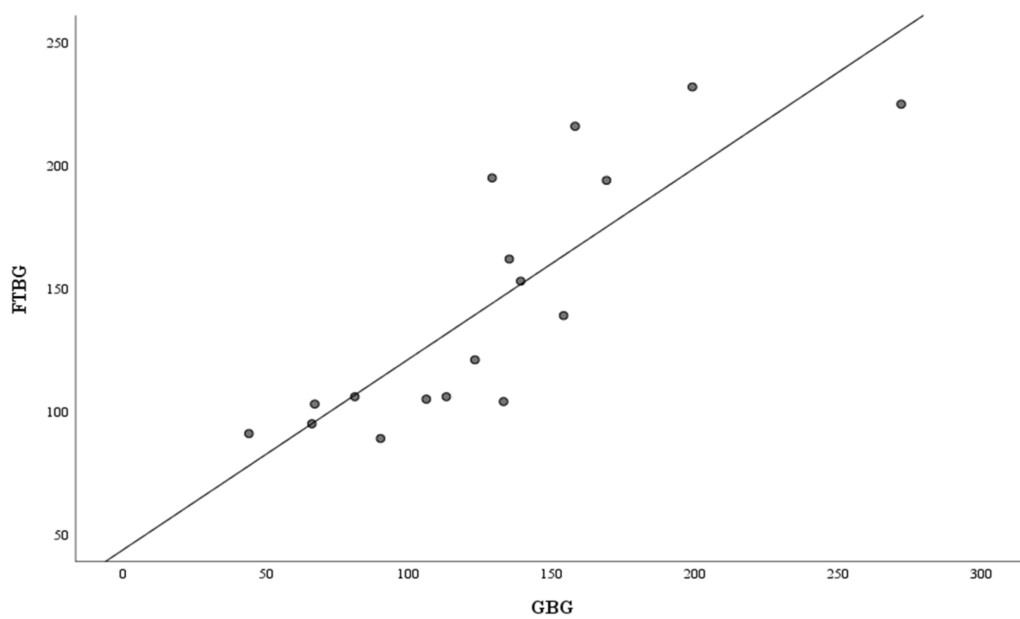

**Figure 8 Correlation and regression analyses for patients with PPD of ≥4 mm.** Linear regression of GBG (mg/dL) sample measurements on FTBG (mg/dL) sample readings for patients with PPD of ≥4 mm. The solid line represents the regression line.

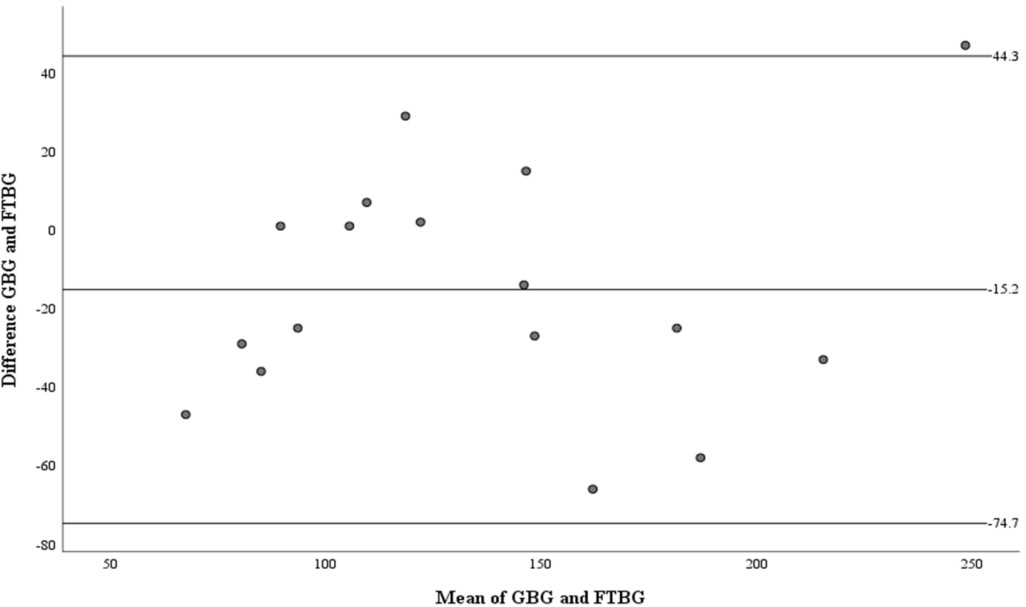

**Figure 9 The Bland-Altman plot for patients with PPD of ≥4 mm.** Means of GBG (mg/dL) and FTBG (mg/dL) glucose readings and differences in GBG (mg/dL) and FTBG (mg/dL) glucose readings, with the lines representing the upper and lower 95% limits of agreement and the mean of the differences for patients with PPD of ≥4 mm.

**Table 4  Subgroup analysis of the repeatability/agreement of the measurements of blood glucose levels (mg/dL) in the GBG and FTBG samples.**

| Measures | Sample site of PPD of ≥4 mm ($n$ =17) | Sample site of PPD of ≤3 mm ($n$ =13) |
|---|---|---|
| Minimum difference | −66 | −73 |
| Maximum difference | 47 | 5 |
| Mean difference ± SD | −15.2 ± 30.4 | −18.9 ± 21.2 |
| 95% CI of mean difference | −30.8 to 0.43 | −31.7 to −6.11 |
| Coefficient of agreement | 59.5 | NA |
| Limits of agreement | −74.7, 44.3 | NA |
| 95% CI of the lower limit | −103 to −46.7 | NA |
| 95% CI of the upper limit | 16.3 to 72.3 | NA |

**Notes:**
GBG, gingival blood glucose; FTBG, fingertip blood glucose; $n$, number; PPD, probing pocket depth; SD, standard deviation; CI, confidence interval; NA, not applicable.

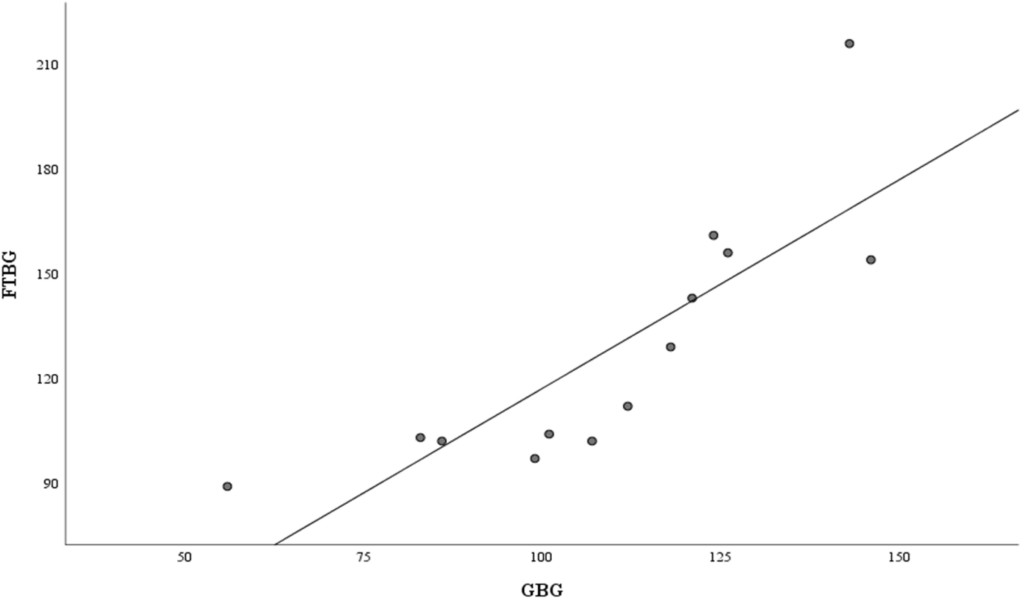

**Figure 10  Correlation and regression analyses for patients with PPD of ≤3 mm.** Linear regression of GBG (mg/dL) sample measurements on FTBG (mg/dL) sample readings for patients with PPD of ≤3 mm. The solid line represents the regression line.

0.823 × GBG value (95% CI for R [0.650 to 1.748], $p$ = 0.001), $R^2$ = 0.677, respectively (Fig. 10).

The Bland–Altman analysis of the PPD ≤3 mm subgroup revealed a significant difference in fixed bias (MD ± SD = −18.9 ± 21.2 mg/dL, 95% CI for MD [−31.7 to −6.11 mg/dL], $p$ = 0.007; Fig. 11 and Table 4). Proportional bias was considered significant (R = −0.774, 95% CI for R [−1.533 to −0.016], $p$ = 0.046).

The AUC of the GBG value was 0.880 (95% CI [0.752 to 1.000], $p$ < 0.001; Fig. 12). The AUC of the FTBG values was 0.802 (95% CI [0.641 to 0.963], $p$ = 0.005; Fig. 12). The sensitivity, specificity, and cutoff values were obtained, as both AUC values were significant. The sensitivity, specificity, and cutoff values of the GBG values were 0.800,

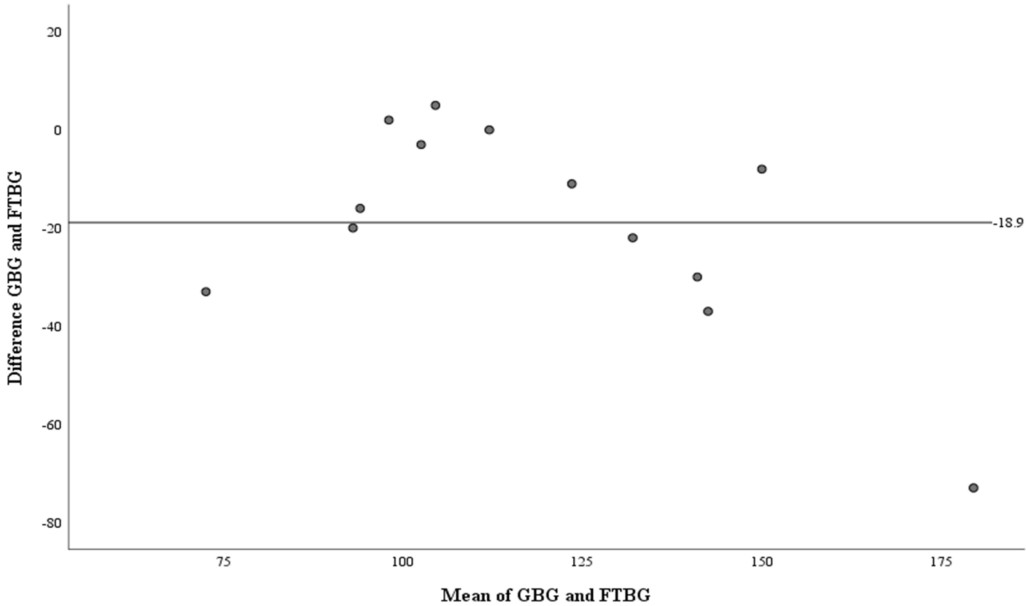

**Figure 11 The Bland-Altman plot for patients with PPD of ≤3 mm.** Means of GBG (mg/dL) and FTBG (mg/dL) levels and differences in GBG (mg/dL) and FTBG (mg/dL) levels, with the line representing the mean of the differences for patients with PPD of ≤3 mm.

0.933, and 123.5 mg/dL, respectively (Table 5). The sensitivity, specificity, and cutoff values of the FTBG values were 0.733, 0.867, and 134.0 mg/dL, respectively (Table 6). The significant difference in AUCs between the GBG and FTBG levels was tested. The difference between the AUCs of the GBG and FTBG values was 0.078 (95% CI [−0.006 to 0.161], $p = 0.068$), which was not significant (Fig. 12).

## DISCUSSION

Periodontal treatment improves glycemic control in patients with type 2 diabetes by improving insulin resistance caused by periodontal inflammation. Therefore, measuring the blood glucose levels in the gingiva of patients with periodontitis may be helpful in screening untreated diabetic patients. Nonetheless, definitive diabetes diagnosis requires HbA1c and an oral glucose tolerance test, limiting the use of GBG to screening. The Bland–Altman analysis revealed a significant difference between all participants' GBG and FTBG levels. Consistent with our findings, the Bland–Altman analysis revealed a significant difference between the GBG and FTBG levels for all participants in a previous report (*Müller & Behbehani, 2005*). The Bland–Altman analysis revealed no significant difference between GBG and FTBG levels in the diabetes or PPD ≥4 mm groups. Similarly, a Bland–Altman analysis revealed no significant difference between the GBG and FTBG levels in a group with PPD ≥4 mm in a previous study (*Strauss et al., 2009*).

The sensitivity, specificity, and cutoff value of the GBG measurements for detecting diabetes were 80%, 93%, and 123.5 mg/dL, respectively. The sensitivity, specificity, and cutoff value of the FTBG measurements for detecting diabetes were 73%, 87%, and 134.0 mg/dL, respectively. No significant differences were observed between the AUCs

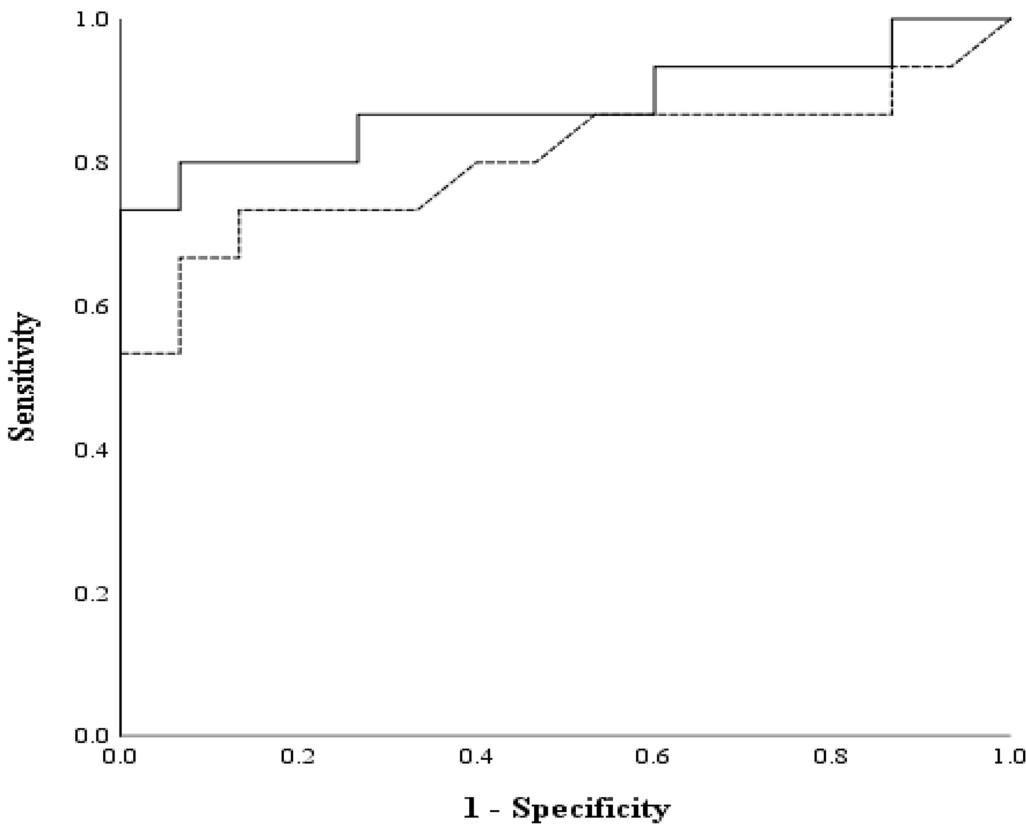

**Figure 12 ROC for the GBG and FTBG levels corresponding to the diagnosis of type 2 diabetes mellitus.** The outer solid line represents GBG, and the inner dashed line represents FTBG.

**Table 5 Coordinates of the ROC curve: GBG cutoff values for diagnosing type 2 DM.**

| Criterion value for a GBG mg/dL cutoff | Sensitivity | 1-Specificity |
|---|---|---|
| 43.0 | 1.000 | 1.000 |
| 50.0 | 1.000 | 0.933 |
| 61.0 | 1.000 | 0.867 |
| 66.5 | 0.933 | 0.867 |
| 74.0 | 0.933 | 0.800 |
| 82.0 | 0.933 | 0.733 |
| 84.5 | 0.933 | 0.667 |
| 88.0 | 0.933 | 0.600 |
| 94.5 | 0.867 | 0.600 |
| 100.0 | 0.867 | 0.533 |
| 103.5 | 0.867 | 0.467 |
| 106.5 | 0.867 | 0.400 |
| 109.5 | 0.867 | 0.333 |
| 112.5 | 0.867 | 0.267 |

| Table 5 (continued) | | |
|---|---|---|
| Criterion value for a GBG mg/dL cutoff | Sensitivity | 1-Specificity |
| 115.5 | 0.800 | 0.267 |
| 119.5 | 0.800 | 0.200 |
| 122.0 | 0.800 | 0.133 |
| 123.5 | 0.800 | 0.067 |
| 125.0 | 0.733 | 0.067 |
| 127.5 | 0.733 | 0.000 |
| 131.0 | 0.667 | 0.000 |
| 134.0 | 0.600 | 0.000 |
| 137.0 | 0.533 | 0.000 |
| 141.0 | 0.467 | 0.000 |
| 144.5 | 0.400 | 0.000 |
| 150.0 | 0.333 | 0.000 |
| 156.0 | 0.267 | 0.000 |
| 163.5 | 0.200 | 0.000 |
| 184.0 | 0.133 | 0.000 |
| 235.5 | 0.067 | 0.000 |
| 273.0 | 0.000 | 0.000 |

Notes:
ROC, receiver operating characteristic; GBG, gingival blood glucose; DM, diabetes mellitus.

| Table 6 Coordinates of the ROC curve: FTBG cutoff values for diagnosing type 2 DM. | | |
|---|---|---|
| Criterion value for an FTBG mg/dL cutoff | Sensitivity | 1-Specificity |
| 88.0 | 1.000 | 1.000 |
| 90.0 | 0.933 | 0.933 |
| 93.0 | 0.933 | 0.867 |
| 96.0 | 0.867 | 0.867 |
| 99.5 | 0.867 | 0.800 |
| 102.5 | 0.867 | 0.667 |
| 103.5 | 0.867 | 0.533 |
| 104.5 | 0.800 | 0.467 |
| 105.5 | 0.800 | 0.400 |
| 109.0 | 0.733 | 0.333 |
| 116.5 | 0.733 | 0.267 |
| 125.0 | 0.733 | 0.200 |
| 134.0 | 0.733 | 0.133 |
| 141.0 | 0.667 | 0.133 |
| 148.0 | 0.667 | 0.067 |
| 153.5 | 0.600 | 0.067 |
| 155.0 | 0.533 | 0.067 |
| 158.5 | 0.533 | 0.000 |

(Continued)

| Table 6 (continued) | | |
| --- | --- | --- |
| Criterion value for an FTBG mg/dL cutoff | Sensitivity | 1-Specificity |
| 161.5 | 0.467 | 0.000 |
| 178.0 | 0.400 | 0.000 |
| 194.5 | 0.333 | 0.000 |
| 205.5 | 0.267 | 0.000 |
| 220.5 | 0.133 | 0.000 |
| 228.5 | 0.067 | 0.000 |
| 233.0 | 0.000 | 0.000 |

Notes:
ROC, receiver operating characteristic; FTBG, fingertip blood glucose; DM, diabetes mellitus.

(0.078, 95% CI [−0.006 to − 0.161]). The specificity and sensitivity of the GBG measurements for detecting diabetes in the present study were comparable with those of previous studies (*Koneru & Tanikonda, 2015*; *Partheeban et al., 2017*; *Sibyl et al., 2017*).

The optimal FTBG level for screening for diabetes is 120–140 mg/dL (*Rolka et al., 2001*; *Zhang et al., 2005*). Furthermore, the follow-up of dental patients with FTBG levels of ≥121 mg/dL detected during screening facilitates the early diagnosis of diabetes (*Engström et al., 2013*). Casual blood glucose monitoring can be performed easily and reliably in dental practice (*Barasch et al., 2013*; *Harase et al., 2015*; *Al-Sebaei et al., 2023*). The optimal fasting GBG level for diabetes screening has been reported to be 125 mg/dL (*Partheeban et al., 2017*); however, the optimal range for casual GBG levels remains unknown. In this study, the cutoff value was determined *via* the ROC curve, and the value at which it was greatest was 123.5 mg/dL. This value was close to the standard for abnormal fasting blood glucose values, but given the sample size of this study, we should be cautious about whether this value is an absolute standard.

To the best of the authors' knowledge, this is the first study to directly compare the GBG levels with the FTBG levels of patients in Japan and examine the accuracy of these diabetes screening methods. Periodontal treatment reportedly improves HbA1c by approximately 0.5% (*Simpson et al., 2022*), and the possibility that the severity of periodontitis may affect blood glucose levels cannot be ruled out. Differences in GBG due to periodontitis status should be explored in future studies. Furthermore, the sample size of 15 participants per group might limit the generalizability of the findings. Future studies with larger sample sizes are necessary to validate these results.

## CONCLUSIONS

This study hypothesizes that measurement of the GBG measurements may be an effective method for screening tool for type 2 diabetes mellitus, as periodontitis tends to be more severe, and BOP tends to be greater in patients with type 2 diabetes mellitus. The GBG and FTBG levels were reliable in the diabetic group and patients with a PPD ≥4 mm. Furthermore, the sensitivity and specificity were as high as those reported in previous studies. Therefore, random blood glucose levels measured from periodontal pockets of patients with diabetes and patients with periodontitis with a PPD of ≥4 mm may be useful

for screening for type 2 diabetes mellitus. These findings are expected to contribute to the early detection of type 2 diabetes through the dental setting.

The limitation of this study is a small sample size that cannot allow us to current data as a standard for the general population. Although measurement using Bland-Altman, correlation, and regression analyses determined the usefulness of the GBG, however, more accurate measurements need to be confirmed using a large number of participants with a variety of clinical conditions. This study suggests that GBG may be useful for screening for diabetes in dentistry. In contrast, there may be limitations regarding the degree of concordance with FTBG. Local factors such as inflammatory cytokines in periodontal tissues may influence GBG, and further investigation of these factors may allow GBG to be applied in diagnosing and treating diabetes-related periodontitis.

## ACKNOWLEDGEMENTS

We appreciate Dr. Nobuyoshi Kitaichi, MD, Ph.D., Director of Health Sciences University of Hokkaido Hospital, for his great cooperation throughout this study. We would like to thank Editage for editing and reviewing this manuscript for English language.

### Funding

This study was supported by Grants-in-Aid from Northern Advancement Center for Science & Technology of Hokkaido Japan (The Development Grants for Medical Institution Needs). The funders had no role in study design, data collection and analysis, decision to publish, or preparation of the manuscript.

### Grant Disclosures

The following grant information was disclosed by the authors:
Grants-in-Aid from Northern Advancement Center for Science & Technology of Hokkaido Japan (The Development Grants for Medical Institution Needs).

### Competing Interests

The authors declare that they have no competing interests.

### Author Contributions

- Yutaka Terada conceived and designed the experiments, performed the experiments, analyzed the data, prepared figures and/or tables, authored or reviewed drafts of the article, and approved the final draft.
- Hiroyuki Watanabe performed the experiments, analyzed the data, prepared figures and/or tables, authored or reviewed drafts of the article, and approved the final draft.
- Mari Mori conceived and designed the experiments, performed the experiments, analyzed the data, prepared figures and/or tables, authored or reviewed drafts of the article, and approved the final draft.
- Kotoko Tomino performed the experiments, prepared figures and/or tables, and approved the final draft.

- Masaya Yamamoto performed the experiments, prepared figures and/or tables, and approved the final draft.
- Mitsuru Moriya conceived and designed the experiments, analyzed the data, authored or reviewed drafts of the article, and approved the final draft.
- Masahiro Tsuji conceived and designed the experiments, analyzed the data, authored or reviewed drafts of the article, and approved the final draft.
- Yasushi Furuichi conceived and designed the experiments, performed the experiments, analyzed the data, authored or reviewed drafts of the article, and approved the final draft.
- Tomofumi Kawakami conceived and designed the experiments, performed the experiments, analyzed the data, authored or reviewed drafts of the article, and approved the final draft.
- Toshiyuki Nagasawa conceived and designed the experiments, performed the experiments, analyzed the data, prepared figures and/or tables, authored or reviewed drafts of the article, and approved the final draft.

### Human Ethics

The following information was supplied relating to ethical approvals (*i.e.*, approving body and any reference numbers):

The Institute of Preventive Medical Science, Health Sciences University of Hokkaido, Ethics Committee granted ethical approval to carry out the study within its facilities (Ethical Application Ref: 2019_028).

### Data Availability

The raw measurements are available in the Supplemental File.

### Supplemental Information

Supplemental information for this article can be found online at http://dx.doi.org/10.7717/peerj.18239#supplemental-information.

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
