# Peer review of "Reliability and utility of blood glucose levels in the periodontal pockets of patients with type 2 diabetes mellitus: a cross-sectional study"

_PeerJ, doi:10.7717/peerj.18239_

## Round 0.1 · original submission · Minor Revisions

Dear authors,

Please refer to the reviewers' comments for further details on the revisions necessary or clarifications that may be requested.

·

Basic reporting

Authors can add a few infographics.
It will be nice if the authors add either one BOX regarding the principal findings of the paper.

Experimental design

Fine

Validity of the findings

This paper complies with validity.

Additional comments

Good Effort
Good Luck

Reviewer 2 ·

Basic reporting

Overall, the article uses clear and professional English, provides sufficient background information and literature references, has a well-organized structure, and shares raw data. However, there are a few areas that need improvement:

Language and Grammar:
Some sentences could be simplified for clarity. For example, the abstract contains some complex sentence structures that can be made more concise.
For instance, “Several studies have reported the measurement of gingival blood glucose (GBG) levels; however, few studies have confirmed the presence of systematic bias using the Bland-Altman analysis.” could be revised to: “Several studies have measured gingival blood glucose (GBG) levels, but few have confirmed systematic bias using Bland-Altman analysis.”
Literature References:
It is suggested to include more recent studies in the background section to ensure the timeliness and relevance of the literature.
For example, when discussing the measurement of GBG, adding references to recent publications would enhance the completeness of the background information.

Experimental design

The experimental design is robust, and the methods are described in detail, but the following areas could be improved:

Sample Size:
The sample size is relatively small (15 participants per group). It is recommended to discuss the impact of the sample size on the study’s findings and to mention in the discussion section whether further research with larger samples is needed to validate the results.
For example: “The sample size of 15 participants per group might limit the generalizability of the findings. Future studies with larger sample sizes are necessary to validate these results.”
Methodological Details:
Some methodological details could be further clarified. For instance, the criteria for selecting participants or the specific procedures for blood glucose measurement should be described in more detail to enhance replicability.
For example: “The inclusion and exclusion criteria for participant selection should be specified more clearly. Additionally, a step-by-step description of the blood glucose measurement procedure would help ensure the study can be accurately replicated.”

Validity of the findings

The study’s findings are robust and statistically sound, but the following points should be considered:

Statistical Analysis:
While the statistical methods used are appropriate, it would be helpful to provide more detailed explanations of the statistical tests and their results in the text.
For example: “A more detailed explanation of the Bland-Altman analysis results, including the interpretation of the bias and limits of agreement, would provide a clearer understanding of the findings.”
Conclusions and Implications:
The conclusions are well-stated and supported by the results, but discussing the broader implications of the findings and potential limitations in more detail would strengthen the article.
For example: “Discussing the potential clinical implications of using gingival blood glucose measurements in dental settings and any limitations related to the study’s scope or methodology would enhance the discussion section.”

Additional comments

The study is commendable for its thorough investigation and clear presentation. The findings are significant for clinical applications in screening type 2 diabetes in dental settings. However, minor revisions are needed to address the above points.

·

Basic reporting

I have reviewed the manuscript titled "Reliability and Utility of Blood Glucose Levels in the Periodontal Pockets of Patients with Type 2 Diabetes Mellitus." The study addresses an important topic and is generally well-structured, with clear aims and comprehensive methodology. However, certain areas require improvement, including detailed reporting of sample size determination, participant selection criteria, and statistical analyses. Additionally, a deeper comparison with existing literature and discussion of the study's limitations would enhance the manuscript. My detailed comments and suggestions are provided below.

For the best interest of the manuscript please consider to add details about the biological mechanisms linking diabetes and periodontal disease, and discuss the advantages and limitations of GBG measurements.
The authors should double-check the citations in the main body of the manuscript, some reference studies are too much old, please consider to update with some lates relevant literature review.
Make sure all the units, formulas and abbreviations used in the manuscript are according to the journal’s guidelines, for this please check the author guideline section of the journal.
I would recommend that please conduct a final proofread to catch minor grammatical errors or awkward phrasing.

Experimental design

Authors, please ensure that the specific inclusion and exclusion criteria were beyond age and dentition status. Were there other health conditions or medications that could influence blood glucose levels and were thus considered?
How recent were the HbA1c measurements used to classify participants into diabetic and non-diabetic groups? Were these measurements taken specifically for the study or obtained from medical records?
The timing of HbA1c measurements can impact their accuracy and relevance to the study period.
Were the periodontal examinations conducted by a single examiner or multiple examiners? If multiple, was there any calibration or inter-examiner reliability assessment performed?
How was the minimum volume of 1.0 μl for blood glucose measurement determined to be sufficient? Was this volume validated against larger sample volumes to ensure accuracy?
Were any adjustments made for potential confounders (e.g., age, sex, smoking status) in the correlation and regression analyses between GBG and FTBG levels?
What criteria were used to define a clinically significant difference between GBG and FTBG levels in the Bland–Altman analysis?
Authors, please consider, how were the cut-off values for diabetes screening determined from the ROC curves? Were these values validated or compared against established clinical thresholds?
How was consistency ensured in the selection of periodontal sites for GBG measurement? Were specific teeth or sites consistently used across all participants?
Be more specific about what future studies should focus on, such as larger sample sizes, different populations, or longitudinal studies.
If manuscript is in English then authors should submit an English version of the ethical statement.

Validity of the findings

How did this study, account for the differences in periodontitis severity between the diabetic and non-diabetic groups in your analysis? Did you control for periodontitis severity in your statistical models?
What criteria did you use to define acceptable limits of agreement in the Bland-Altman analysis? How do these criteria relate to clinical relevance?
Did you perform any cross-validation or external validation of the cut-off values derived from the ROC analysis? How robust are these cut-off values across different populations?

Additional comments

In the discussion section, please, emphasize the potential clinical applications of using GBG measurements and compare your findings with other studies to highlight consistencies and discrepancies.
Discuss how the small sample size and differences in periodontitis severity might impact the generalizability of the findings.

---

## Round 0.2 · Minor Revisions

Dear authors,

We still need some minor revisions before acceptance. I highlight the consistency in terminology use. But I also find it relevant to justify the statistical methods approach and also mention limitations more clearly.

·

Basic reporting

Good can be accepted.

Experimental design

Good.

Validity of the findings

It is fine.

Additional comments

This paper can be accepted if it satisfies publisher policy.

Reviewer 2 ·

Basic reporting

The manuscript is generally well-written in clear, professional English. The introduction and background provide sufficient context and the literature is appropriately referenced. The article structure conforms to standards, with relevant figures and tables included. Raw data has been provided. The results align with the stated hypotheses.

Suggested improvements:

Some grammatical errors should be corrected, for example in the Abstract: "Pearsonís -B correlation" and "BlandñAltman"
Slightly more background information in the Introduction on the rationale for comparing gingival blood glucose to fingertip glucose would be helpful
Statistical tests should be named consistently - both "BlandñAltman analysis" and "=B analysis" are used
The Conclusions section in the Abstract could be expanded slightly

Experimental design

This study addresses a well-defined and relevant research question regarding the reliability of gingival blood glucose compared to fingertip blood glucose in patients with diabetes. The methods are described in sufficient detail to replicate, including inclusion/exclusion criteria, sample size calculation, and examination procedures. The study appears to have been conducted to high technical and ethical standards.

Validity of the findings

The authors have provided all the underlying data, which appear robust and controlled. Statistical analyses are sound. The conclusions are stated clearly and align with the original research question and results.

Suggested improvements:

It would be helpful to briefly explain why Bland-Altman analysis was used in addition to correlation and regression
Limitations of the study should be acknowledged more explicitly in the Conclusions

·

Basic reporting

The authors have successfully addressed all the concerns raised in the initial review of their manuscript titled "Reliability and Utility of Blood Glucose Levels in the Periodontal Pockets of Patients with Type 2 Diabetes Mellitus: A Cross-Sectional Study." The revisions have improved the clarity, organization, and scientific stringency of the study. The methods and results are now well-presented, with appropriate approach, and the discussion effectively contextualizes the findings within the existing literature. Ethical considerations have been clearly outlined, and the language has been polished for better readability. Overall, the manuscript is now suitable for publication.

Experimental design

There are no specifical comments for this section.

Validity of the findings

I have no further questions.

---

## Round 0.3 · accepted · Accept

Dear authors,
i am pleased to let you know that I am accepting your manuscript for publication. Many congratulations and thank you.